# Enhancing Laying Hens’ Performance, Egg Quality, Shelf Life during Storage, and Blood Biochemistry with *Spirulina platensis* Supplementation

**DOI:** 10.3390/vetsci11080383

**Published:** 2024-08-21

**Authors:** Md Salahuddin, Ahmed A. A. Abdel-Wareth, Kayla G. Stamps, Cassandra D. Gray, Adrian M. W. Aviña, Sadanand Fulzele, Jayant Lohakare

**Affiliations:** 1Poultry Center, Cooperative Agricultural Research Center, Prairie View A&M University, Prairie View, TX 77446, USA; mdsalahuddin@pvamu.edu (M.S.); or a.wareth@agr.svu.edu.eg (A.A.A.A.-W.); kstamps2@pvamu.edu (K.G.S.); cdgray@pvamu.edu (C.D.G.); amavina@pvamu.edu (A.M.W.A.); 2Department of Animal and Poultry Production, Faculty of Agriculture, South Valley University, Qena 83523, Egypt; 3Department of Neuroscience and Regenerative Medicine, Medical College of Georgia, Augusta University, Augusta, GA 30912, USA; sfulzele@augusta.edu

**Keywords:** egg quality, health status, laying hens, production, microalgae, serum biochemistry

## Abstract

**Simple Summary:**

This present study was carried out to investigate the benefits of introducing *Spirulina platensis* (SP), a highly nutritional microalgae, into the diets of laying hens. Our objective was to investigate the impact of varying quantities of SP (2.5 g/kg, 5 g/kg, and 10 g/kg) on egg production, egg quality, and the overall health of the laying hens. During a period of six weeks, we observed significant enhancements in the weight and mass of eggs from chickens that were given SP. Additionally, these hens exhibited darker yolk color and higher quality albumen, which are indicative of eggs’ freshness. The addition of SP to hens’ diet improved their liver function and elevated the protein level in their blood serum. Furthermore, the eggs from these hens exhibited improved shelf life in terms of quality stability during storage for 21 days. The findings indicate that introducing 2.5 g/kg and 5 g/kg of SP into the diet of hens can enhance productivity, boost egg quality and shelf life, and contribute to better health, thus increasing the sustainability and efficiency of chicken farming. This study emphasizes the potential of SP as a beneficial supplement for laying hens, which can enhance food production for society.

**Abstract:**

Enhancing the sustainability of chicken farming involves improving health and productivity and product qualities. This study explores the influence of *Spirulina platensis* (SP) supplementation on the productivity, egg quality, shelf life during storage, and blood biochemistry of laying hens. A total of 192 thirty-nine-week-old White Leghorn hens were randomly divided into 4 dietary groups: a control group and 3 treatment groups receiving 2.5 g/kg, 5 g/kg, or 10 g/kg of SP, respectively. The study was conducted for six weeks with measuring feed intake, feed conversion ratio, egg production, egg quality, shelf life, and blood biochemistry. The results demonstrated significant enhancements in egg weight (*p* < 0.05) and egg mass (*p* < 0.05) in the treatment of SP groups. The SP treated hens showed significant improvements in yolk color (*p* < 0.05) and Haugh unit scores (*p* < 0.05). The SP supplementation showed a hepatoprotective effect, as indicated by significant reduction in Alanine aminotransferase (ALT) (*p* < 0.05) and alkaline phosphatase (ALP) (*p* < 0.05) levels; however, increases in total protein, albumin, and globulin levels were observed. Furthermore, the egg quality of stored eggs for 21 days linearly increased with increments in the SP levels. In conclusion, it can be speculated that adding SP at 2.5 g/kg and 5 g/kg can significantly improve the productivity of laying hens, eggs’ quality, shelf life, and blood biochemistry, thereby contributing to a more sustainable and efficient chicken production.

## 1. Introduction

Poultry farming is crucial for meeting global demand for animal protein, and enhancing laying hens’ health and efficiency is essential for sustainability. Conventional feed ingredients primarily fulfill nutritional needs [1], but there is growing interest in functional feed additives. In recent years, algae, such as *Spirulina platensis* (SP), have gained attention for their potential to enhance poultry performance and health. Algae are photosynthetic organisms found in aquatic environments, ranging from single-celled microalgae to large seaweeds [2].

*Spirulina platensis* (SP), commonly referred to as Spirulina, is a blue-green microalga known for its high concentration of protein, essential amino acids, vitamins, minerals, and bioactive substances [3,4]. SP is rich in vitamin B12, γ-linolenic fatty acid, β-carotene, calcium, and iron [3,4], making it an attractive option for supplementing animal diets. Vitamin B12 is crucial for red blood cell formation and immunological function, while γ-linolenic fatty acid helps maintain cell membrane integrity and produces anti-inflammatory eicosanoids [5,6]. β-carotene, a precursor to vitamin A, is vital for vision, immune function, and skin health [7]. Calcium is essential for bone health, ensuring the proper development and maintenance of the skeletal system, while iron is critical for the transport of oxygen in the blood, supporting overall metabolic functions [8,9]. Beyond these essential nutrients, SP is packed with bioactive compounds that offer various health benefits. Phycocyanin, a pigment–protein complex found in SP, has potent antioxidant and anti-inflammatory properties. It helps neutralize free radicals, thereby protecting cells from oxidative damage and reducing inflammation [10]. The polysaccharides present in SP also contribute significantly to its health-promoting effects. These complex carbohydrates enhance immune function by stimulating the activity of macrophages, natural killer cells, and other components of the immune system [11]. Together, these bioactive compounds support overall health and productivity in poultry by improving their immune response, reducing oxidative stress, and promoting better nutrient utilization. The distinctive nutritional composition of Spirulina makes it a highly attractive option for supplementing animal diets. Based on various research, SP has been found to enhance productive performance and improve the overall health of ruminant and non-ruminant animals [12,13,14,15,16,17].

Research studies have demonstrated that the dietary utilization of SP exhibits beneficial effects on poultry species. Elsewhere, studies found that regular protein sources in broiler diets could be replaced by up to 15% SP without any adverse impact on meat quality and productivity [18,19,20]. Furthermore, a recent study conducted by El-Hady et al. [21] found that the inclusion of 6% SP resulted in enhanced growth performance and an improved gut health condition in broiler chickens compared to 0% SP. In addition, studies have demonstrated that adding 1% SP to broiler diets can effectively counteract the detrimental impacts of aflatoxins on performance, immunological function, blood biochemical features, and cecal microbiota [22,23].

On the other hand, supplementing the laying hens’ diet with 0.3% SP did not result in any changes in egg production, egg weight, egg mass, feed intake (FI), feed conversion ratio (FCR), Haugh unit, yolk color, percentages of eggshell, albumen, yolk, or certain biochemical parameters [24,25]. The inclusion of up to 2% SP in the laying hens’ diet led to no improvement in FI, egg production, egg weight, egg mass, or FCR, but did result in enhanced eggshell thickness, strength, and yolk color [26]. Another study demonstrated that a diet supplemented with 2% SP had no significant effect on egg production, egg mass, egg weight, FI, FCR, shell strength, shell thickness, albumen index, and serum parameters [27]. Conversely, a six-week study with 2.5% SP supplementation showed higher egg weight and darker yolk color but a lower Haugh unit score [28]. Other similar research suggested that the inclusion of SP up to 2.5% in the diets of laying hens solely improved egg yolk color, without significant effects on egg production, FI, FCR, and egg weight [29]. In addition, a study including 3% SP in the laying hens’ diet found significant alterations in egg lipid profile and yolk color [30]. These observed variations in the impact of SP on laying hens may be due to differences in the levels of SP supplementation used in various studies, which could influence the overall effectiveness of SP in enhancing hen performance. Therefore, synthesizing the reasons behind these varying impacts, the primary objective of this study is to determine the optimal levels of SP supplementation that can maximize the benefits on laying hen performance. Specifically, this study aims to identify the levels of SP that most effectively improve egg production rates, enhance egg quality parameters such as the Haugh unit score, eggshell strength, thickness, and yolk color, as well as extending the shelf life of eggs during storage by maintaining their freshness and quality over time. Additionally, the study seeks to understand how SP supplementation can positively influence blood biochemistry, thereby supporting the overall health, immune function, and metabolic efficiency of laying hens.

To achieve this, we meticulously selected inclusion rates of 2.5, 5, and 10 g/kg SP. These rates were chosen based on a comprehensive review of the existing literature that indicated these levels might provide a range of effects from minimal to substantial. Identifying the SP optimal supplementation levels aims to promote better health and productivity in laying hens, thus contributing to more sustainable and efficient egg production practices.

## 2. Materials and Methods

The present study was conducted at the poultry center at Prairie View A&M University, Prairie View, TX, USA, following the Institutional Animal Care guidelines and regulations. The approval number for this experiment is AUP# 2023-046.

### 2.1. Animals, Experimental Design and Feed Management

In the experiment, a total of 192 thirty-nine-week-old White Leghorn laying hens, with an average body weight of 1676.34 ± 29.58 g, were used. Before the start of the experiment, hens were fed a diet formulated to meet the nutrient requirement of White Leghorn laying hens, containing 2882 kcal/kg of AME and 165 g/kg of crude protein (Table 1) which was formulated in accordance with the guidelines provided by the National Research Council [31]. The hens were randomly allocated to four dietary treatment groups: a control group receiving a standard diet without *Spirulina platensis* (SP), and three experimental groups receiving the standard diet supplemented with 2.5 g/kg, 5 g/kg, and 10 g/kg SP, respectively. Each group consisted of eight replicates, with six hens per replicate. Experimental diets were provided after completing a 7-day adaptation period and the study was carried out for a six-week duration. The SP was provided for free by Earthrise Nutritionals LLC, Calipatria, CA, USA. The proximate analysis of SP revealed a high protein content (69.3 g/100 g), both essential and non-essential amino acids, substantial mineral quantities, and a variety of phytonutrients, including c-phycocyanin, β-carotene (0.23 g/100 g), and zeaxanthin (0.15 g/100 g). The hens were housed on the floor with wood shavings bedding in a semi-open house. All groups were allowed ad libitum access to feed and water throughout the experimental period. Feed was supplied weekly, and the remaining feed was measured prior to each refill to ensure the accurate monitoring of feed intake. Eggs were collected daily, then counted and weighed for each replicate within each group to measure the egg production, feed conversion ratio, and quality.

### 2.2. Hens’ Performance and Egg Production Parameters

Weekly feed intake (FI) was calculated by subtracting the remaining feed from the amount initially provided. The hen day egg production (HDEP) rate was calculated by dividing the total number of eggs laid per hen each day by the number of hens and expressing it as a daily percentage. The weight of the eggs was measured every day, and the mass of each egg was determined using the formula: egg mass (g/hen/day) = (average egg weight (g) × hen day egg production (%))/100. The feed conversion ratio (FCR) was determined by dividing the feed intake (g) by the egg mass (g). Production measurements, including feed intake and egg production, were adjusted to consider hen mortalities, with daily recording of deaths as they happened. These modifications maintained the accuracy of the production data for each replication, which was subsequently calculated for each hen to precisely represent the actual performance metrics.

### 2.3. Determination of Egg Quality Parameters

Egg internal quality assessments were conducted every two weeks during the experiment and again three weeks after the experiment concluded to evaluate the quality of stored eggs. At the end of the experiment and three weeks later, external quality measurements were conducted, with particular emphasis on egg composition. A total of twenty-four eggs were picked at random from each treatment group, with three eggs per replication. The egg quality parameters were analyzed using the Egg Tester Ultimate™ (12-ETU-001 Version B, ORKA Food Technology Ltd., West Bountiful, UT, USA). The Haugh unit, a measure of egg albumen quality, was measured as a score, calculated based on the height of the egg white and the weight of the egg. Egg strength was measured in kilograms (kg). Yolk color was measured as a score, and shell thickness was measured in millimeters (mm). The weight of each egg was measured separately in grams (g). After weighing, the eggs were cracked onto a plate and evaluated using the Egg Tester Ultimate™. The yolk was then isolated from the albumen, and the weight of each component was measured in grams (g). To calculate the percentage of the shell, albumen, and yolk in relation to the total egg weight, the weight of each component was divided by the weight of the whole egg and then multiplied by 100.

### 2.4. Blood Sampling and Laboratory Analysis

At the end of the experimental period, sixteen birds from each treatment group were randomly selected, with two birds selected that represented the pen from each replicate. A volume of 5 mL of blood was collected from each hen from the wing vein using sterilized syringes to analyze various serum biochemical parameters. The hens were not subjected to feed restriction prior to blood sampling. After allowing the serum to separate naturally, it was centrifuged for 10 min at 3000× *g* using a Sorvall ST Plus centrifuge (Thermo Electron LED GmbH, Langenselbold, Germany). The serum samples were then collected using a pipette and transferred into Eppendorf tubes for further analysis. The concentrations of various serum components such as sodium, chloride, calcium, phosphorus, glucose, cholesterol, amylase, uric acid, blood urea nitrogen (BUN), total bilirubin, alanine aminotransferase (ALT), gamma-glutamyl transferase (GGT), alkaline phosphatase (ALP), total protein, albumin, and globulin were determined by an external laboratory, the Arkansas State Veterinary Laboratory in Little Rock, AR, USA.

### 2.5. Statistical Analysis

The normal distribution of the data was determined using the Kolmogorov–Smirnov test, employing the Data Analysis Toolpak add-in in Excel 2019, Version 16. The statistical analysis was conducted using a completely randomized design and the general linear models’ procedure of SAS 9.2 (SAS Institute, 2009). The model included only the level of supplementation. Orthogonal polynomial contrasts were utilized to determine the linear and quadratic effects of increasing supplementation levels. Additionally, Duncan’s multiple range test was used to compare treatment means, with significance declared at *p* < 0.05. *p*-values less than 0.001 were reported as “<0.001” instead of the exact value. The figures were generated by using GraphPad Prism software, version 9 (GraphPad Software, La Jolla, CA, USA).

## 3. Results

### 3.1. The Effect of Spirulina platensis on the Productive Performance of Laying Hens

The influence of *Spirulina platensis* (SP) on the productive performance of laying hens in different age groups (39–44 weeks) yielded diverse effects, as shown in Table 2. Between the ages of 39 and 40 weeks, there were no significant differences in egg weight, egg mass, HDEP, FI, and FCR among the groups. In the age group of 41–42 weeks, the egg weight (*p* = 0.023) and egg mass (*p* = 0.013) were quadratically higher in the groups that received 2.5 and 5 g/kg of SP compared to control group. However, there were no significant changes observed in HDEP, FI, and FCR among groups. In the age group of 43–44 weeks, the treatment groups exhibited a significant quadratic increase in egg weight (*p* = 0.045) and egg mass (*p* = 0.006) when compared to the control group. Furthermore, there were linear (*p* = 0.041) and quadratic (*p* = 0.015) increments in feed intake in the SP groups compared to the control groups. Overall, during the age period of 39–44 weeks, the 2.5 and 5 g/kg groups showed a quadratic greater egg weight (*p* = 0.042) and egg mass (*p* = 0.005) compared to the control group. However, there were no significant differences observed in HDEP, FI, and FCR across the groups.

### 3.2. The Effect of Spirulina platensis on Egg Quality Criteria of Laying Hens

The effect of SP on the egg quality parameters of laying hens showed a change among different age groups (Table 3). At 40 weeks, the group that received 5 g/kg had a quadratic higher Haugh unit score (*p* = 0.022) compared to control group, and the groups that received 2.5 and 5 g/kg of SP showed significantly higher yolk color score with both linear (*p* < 0.001) and quadratic (*p* = 0.036) effects. However, no differences were observed in egg strength and shell thickness across groups. At 42 weeks, the groups receiving 5 and 10 g/kg showed a linear (*p* < 0.001) increase in the yolk color score compared to the control and 2.5 g/kg groups. However, no differences were observed in the Haugh unit score, egg strength, and shell thickness between the control and treatment groups. At 44 weeks, the treatment groups exhibited a linearly (*p* < 0.001) significant increase in yolk color score compared to the control group. The quadratic (*p* = 0.032) significance resulted in higher egg strength in the treatments compared to the control. Moreover, the treatment groups that received 2.5 and 5 g/kg SP showed quadratic (*p* = 0.028) lower shell percentage than the control group. However, there were no observed significant changes in the Haugh unit score, shell thickness, albumen percentage, and yolk percentage among groups.

### 3.3. Effects of Spirulina platensis on Storage Egg Quality

The effects of different levels of SP on various measures of egg quality for stored eggs are summarized in Table 4. There were no significant differences in egg weight among the groups. However, the Haugh unit score showed a significant linear (*p* = 0.050) improvement in the group receiving 5 g/kg SP compared to the control and 2.5 g/kg SP groups. The treatment groups showed linear (*p* = 0.009) increases in yolk color score, and linear (*p* = 0.005) and quadratic (*p* = 0.035) increases in egg strength compared to the control group. The groups receiving 2.5 and 5 g/kg exhibited a quadratic (*p* = 0.006) increase in shell thickness compared to the control group. Additionally, the 10 g/kg group showed a linear rise in shell percentage (*p* = 0.01) and in yolk percentage (*p* = 0.019) compared to the control group while also showing a significant linear reduction in albumen percentage (*p* = 0.003).

### 3.4. The Effects of Spirulina platensis on Serum Biochemical Parameters of Laying Hens

The effects of SP on the serum minerals levels of laying chickens are presented in Figure 1. There were no significant differences in the serum sodium (a), chloride (b), calcium (c), and phosphorus (d) concentrations between control and treatment groups. Figure 2 shows the nutritional biomarker’s concentration in blood serum. No significant differences were noticed in the serum glucose (a), cholesterol (b), and amylase (c) levels among groups. Figure 3 illustrates the effects of different treatments on the liver enzyme levels of GGT (a), ALT (b), and ALP (c). There was linear reduction in serum ALT (*p* = 0.006) and ALP (*p* = 0.011) in the treatment groups compared to the control group. No significant changes in GGT levels were observed between the treatment and the control groups. Figure 4 shows the concentrations of uric acid (a), BUN (b), and total bilirubin (c). No significant differences were observed in uric acid, BUN, and total bilirubin levels between the control and treatment groups. The levels of total protein (a), albumin (b), globulin (c), and the albumin/globulin ratio (d) are shown in Figure 5. The treatment groups exhibited a linear (*p* = 0.024) and quadratic (*p* = 0.029) elevation in total protein levels compared to the control group. The 10 mg/kg SP group showed a significant linear (*p* = 0.009) increase in albumin levels compared to the control group. On the other hand, the 2.5 and 5.0 mg/kg groups showed a quadratic (*p* = 0.040) significant increase in globulin levels compared to the control group. However, there were no changes in the albumin/globulin ratio among groups.

## 4. Discussion

The results of the present study demonstrate that the incorporation of SP into the diet of laying hens has diverse impacts on their productivity, egg quality, egg shelf life, and serum biochemical parameters. These effects differ depending on the levels of the SP supplementation in the diets.

The results revealed no significant variations in egg weight, egg mass, HDEP, FI, and FCR between the control and treatment groups during the age period of 39–40 weeks. This preliminary observation indicates that at the beginning of supplementation, SP does not have an immediate effect on these fundamental performance measurements. This may suggest that the physiology of hens to adapt to the SP supplementation may need a longer period to show significant improvements in productive performance parameters. However, as the investigation advanced to the age groups of 41–42 weeks and 43–44 weeks, an obvious improvement pattern became evident. The hens that were given 2.5 g/kg and 5 g/kg of SP showed significant rises in both egg weight and egg mass compared to the control group and 10 g/kg SP group. More specifically, these increases followed a quadratic pattern, suggesting that there is a relationship between the level of SP supplementation and the performance improvements. Conversely, the reduced egg weight and egg mass observed in birds given 10 g/kg SP during 41–42 and 43–44 weeks of age suggest that higher levels of SP supplementation may not be beneficial for these parameters in these ages of laying hens. This finding indicates that while moderate SP levels (2.5 g/kg and 5 g/kg) are effective, the 10 g/kg level may be excessive and counterproductive. It highlights the importance of optimizing SP inclusion levels to balance the benefits and potential drawbacks. Therefore, higher levels should be approached with caution.

However, the significant improvement in egg weight and egg mass observed during the mid-stages of egg-laying hens with 2.5 g/kg and 5 g/kg SP supplementation indicates that the incorporation of SP into the diet has a delayed but significant beneficial effect on the laying hens. The delayed response can be attributed to the cumulative effects of SP’s rich nutrient profile, which includes high concentrations of protein, essential amino acids, vitamins, minerals, and bioactive compounds such as β-carotene and γ-linolenic acid [3,4]. The nutrients in SP might have stimulated the ovaries and promoted the development of larger eggs, which is particularly beneficial for egg producers seeking to meet specific market size requirements. Similarly, the diverse benefits of algae, including their antioxidant and antimicrobial properties, can positively impact productive performance and egg production [2]. These components likely enhance the metabolic efficiency and overall health of the hens over time. These findings are consistent with previous research indicating that SP can improve the laying performance of hens when incorporated into their diets at appropriate levels. A study conducted by Selim et al. [25] found that groups receiving a 0.3% SP supplementation exhibited a trend to produce heavier eggs and higher egg mass. In a similar vein, Ahmadi Nia et al. [32] observed that incorporating a diet supplemented with 0.6% SP resulted in a significant increase in both egg mass and productivity. In addition, Panaite et al. [33] noticed that incorporating 2% SP into the diet could increase the egg weight. Based on these data, the current study proposes that SP could be used as a beneficial feed supplement to enhance the laying performance of hens.

The incorporation of SP showed significant benefits on numerous egg quality indices throughout the duration of the trial. At 40 weeks, hens that were given a 5 g/kg SP showed a significantly higher Haugh unit score in comparison to the control group, suggesting an enhancement in the quality of the egg albumen. The Haugh unit is a well-established metric for assessing the quality of eggs, which is determined by measuring the height of the albumen. This measurement indicates the freshness and overall quality of the egg [34,35,36]. Some studies have demonstrated that the Haugh unit score was considerably greater in laying hens that were given a diet containing SP compared to those that were not given any supplementation in their diet [24,28,37]. The increased Haugh unit scores found in the group supplemented with SP indicate that SP may improve the protein content and structural integrity of the egg albumen, resulting in enhanced overall egg quality. However, in the subsequent weeks (42 and 44 weeks), there were no significant differences in the Haugh unit scores among the four groups. The lack of persistent improvement in the Haugh unit indicates the positive effect of SP on albumen quality is immediate but not maintained over time. The initially observed improvement in the Haugh unit resulting from SP supplementation may diminish as the hens’ metabolism adapts to the dietary adjustment over time, or other metabolic mechanisms may involve stabilizing albumen quality in the long run.

Moreover, both the groups that received 2.5 g/kg and 5 g/kg of SP showed enhanced yolk color at 40 weeks, along with increased albumen quality. This increase in yolk pigmentation corresponds with findings from other research that have indicated the significant impacts of SP supplementation on egg yolk color. Tufarelli et al. [26] and Khan et al. [24] found that introducing SP to hens’ diets enhances yolk coloration. In a similar vein, a recent study conducted by Panaite et al. [33] demonstrated that incorporating SP supplements into the diet can improve the color of egg yolks. The improved yolk coloration is most likely a result of the high concentration of carotenoids and other pigments in SP [38], which contribute to the development of visibly attractive egg yolks that are often chosen by customers. The improvements in yolk color continued at 42 weeks, where the groups receiving 5 g/kg and 10 g/kg SP showed a linearly significant increase in yolk color compared to the control and 2.5 g/kg groups. These findings indicate that the effect of SP on yolk pigmentation is influenced by the dosage, with greater levels leading to more obvious improvements in color. The continued improvement in yolk color at 44 weeks further confirms the continuing influence of SP administration over a period of time. This phenomenon suggests that the groups receiving SP continuously might have the long-term benefits of SP on egg yolk pigmentation.

In addition, the current study for the groups that received SP supplementation showed an increase in egg strength at 44 weeks. The strength of eggshells is an important requirement because stronger shells decrease the chances of breakage when they are being handled or transported [39,40]. The high mineral content of SP, specifically calcium and iron, is believed to be a contributing factor to these enhancements. Calcium is necessary for developing a strong eggshell structure, while iron contributes to multiple physiological processes that maintain the integrity of the eggshell [41,42].

The eggshell percentage in the 2.5 g/kg and 5 g/kg SP groups was reduced in comparison to the control group, indicating a more effective distribution of calcium and other minerals throughout the eggshell. Additionally, when the SP supplementation was 10 g/kg in laying hens’ diet, the percentage of eggshells exhibited a rising trend, reaching levels similar to those in the control group. Nevertheless, higher levels of SP (10 g/kg) lead to a surge in eggshell percentages. Eggshell strength and quality are essential for preventing breakage during handling and transportation. These improvements may be due to SP supplements, rich in nutrients like calcium and phosphorus, which significantly enhance eggshell quality by supporting strong shell formation [2]. Additionally, spirulina-based diets have been shown to significantly improve average egg weight, yolk color, eggshell strength, and iron content [43]. Our results are consistent with the results reported by Dogan et al. [44] and Samia et al. [45], who observed a similar effect of higher SP supplementation on shell percentages. These data emphasize the intricate nature of the association between SP supplementation and the quality of eggshells. The observed quadratic decline and subsequent rise in shell percentage with increasing SP levels indicate that adjusting the SP level is essential for achieving an optimal proportion between shell strength and shell percentage. Further research is necessary to develop a comprehensive understanding of the basic mechanisms involved and to establish the optimum levels of SP supplementation that can effectively maintain or improve eggshell quality while not negatively impacting other crucial egg characteristics.

Surprisingly, the present study found that eggs from hens that were given SP supplementation could maintain high quality even after being stored. The group that received 5 g/kg of SP showed a significant linear enhancement in Haugh unit scores compared to both the control group and the group that received 2.5 g/kg of SP. This suggests an improvement in the quality of the egg albumen and more effective preservation of egg freshness over time. This enhancement implies that SP has the potential to maintain the structural integrity and thickness of the albumen, hence ensuring its stability during storage. In addition, the supplementation of SP led to enhancements in yolk color and egg strength. The rich carotenoids and pigments found in SP are expected to improve the coloration of the yolk, while its high mineral content, such as calcium and iron, contributes to the development of the eggshell [3,4,41,42]. Furthermore, the groups receiving 2.5 g/kg and 5 g/kg of SP showed a quadratic increase in shell thickness compared to the control group. This indicates improved structural integrity during storage. The group receiving 10 g/kg of SP demonstrated a linear increase in the percentages of shell and yolk, hence improving the eggs’ nutritional content and eggshell strength [45,46]. The results suggest that the addition of SP supplements may effectively preserve the quality of eggs during storage by enhancing the stability of the albumen, pigmentation of the yolk, and strength of the shell. Therefore, SP supplements are a significant dietary addition in poultry nutrition to prolong the egg shelf life.

Serum biochemical parameters analysis exhibited a wide spectrum of trends and also indicated distinct patterns. There were no significant differences in serum sodium, chloride, calcium, and phosphorus levels between the control group and the treatment groups. These findings indicate that SP does not have a negative impact on the mineral balance in laying hens, which is essential for maintaining overall physiological homeostasis.

No significant differences were observed in glucose, cholesterol, and amylase levels across the groups regarding nutritional biomarkers. This finding suggests that the addition of SP does not have an adverse impact on metabolic health or the functioning of digestive enzymes, therefore confirming its safety as a dietary additive. However, this study demonstrated significant effects on liver enzymes. As the concentration of SP increased, there was a significant decrease in ALT levels. Similarly, ALP levels exhibited a linear decrease. The decrease in liver enzyme levels indicates that SP may have protective effects on the liver, which is consistent with previous studies conducted on chickens [25,26,47,48]. The hepatoprotective effects reported may be related to the bioactive compounds found in SP, especially phycocyanin and polysaccharides, which have demonstrated the ability to improve liver function and prevent liver damage [49,50]. Moreover, there were no significant alterations in GGT levels between the control and treatment groups, providing further evidence that SP supplementation does not have negative impacts on liver function. Furthermore, the study revealed that there were no significant differences in serum uric acid, BUN, and total bilirubin concentrations, suggesting that the administration of SP does not have an impact on kidney functions. The findings are consistent with previous studies that have shown the safety of SP as a dietary additive, with no negative impact on kidney functions [51,52].

Protein-related biomarkers examination revealed that the treatment groups had significantly higher overall protein levels compared to the control group. This increase was observed in both a linear and quadratic manner, implying that the addition of SP may enhance protein synthesis and improve the overall nutritional condition of laying hens. Increased protein levels have a vital role in various fundamental biological activities, such as the synthesis of enzymes, hormones, and antibodies, as well as tissue regeneration and development [53,54]. Particularly, the group that received 10 g/kg of SP had a linear increase in albumin levels, which suggests an improvement in liver function and overall metabolic well-being. In addition, the groups that received 2.5 g/kg and 5 g/kg of SP showed a quadratic increase in globulin levels, indicating enhanced immune function and the hens’ capacity to combat infections. These findings align with previous research that has shown that SP has a beneficial effect on protein metabolism and immunological function [55,56,57]. Additional research is recommended to investigate the persistent consequences of SP and to establish the most effective levels of supplementation for maximizing these advantages.

## 5. Conclusions

The results of this study provide evidence that the addition of *Spirulina platensis* (SP) to the diet of laying hens has a significant impact. SP, especially at levels of 2.5 g/kg and 5 g/kg, enhanced egg weight and egg mass and enhanced the quality of albumen, yolk color, and eggshell strength. Furthermore, SP exhibited hepatoprotective effects and promoted protein synthesis without any adverse effects on metabolic health or kidney functions. Significantly, the addition of SP also effectively maintained the quality of eggs during storage, by preserving the integrity of the albumen and improving the coloration of the yolk and the strength of the shell. Therefore, based on our findings, the optimum inclusion levels of SP for laying hens are 2.5 g/kg and 5 g/kg. These levels provided significant improvements in egg production and quality during the mid-stages of laying. In conclusion, adding SP as a dietary supplement enhances productivity, egg quality, egg shelf life, and the health of laying hens, thereby allowing more sustainable and efficient poultry production.

## Figures and Tables

**Figure 1 vetsci-11-00383-f001:**
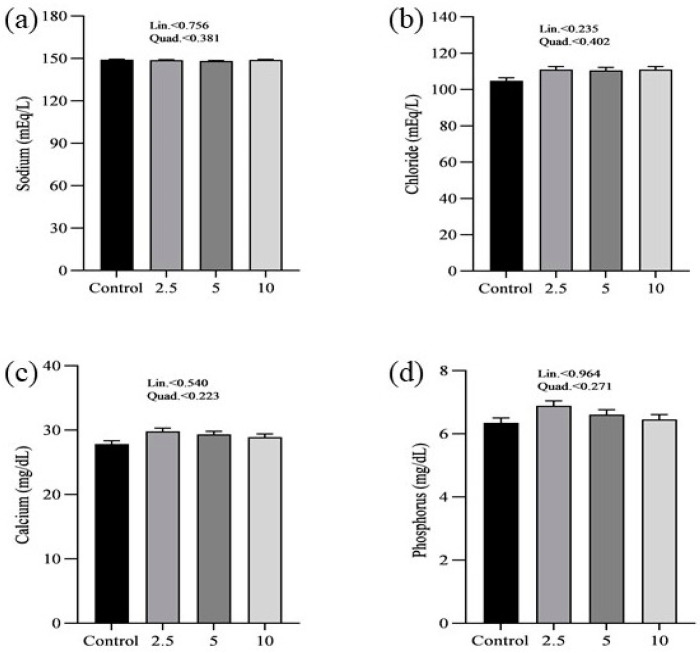
Effect of *Spirulina platensis* supplementation on serum mineral levels in laying hens. Data are expressed as mean ± SEM for each group (n = 16). Panels show the levels of different minerals: (**a**) sodium (mEq/L), (**b**) chloride (mEq/L), (**c**) calcium (mg/dL), and (**d**) phosphorus (mg/dL). The control group did not receive *Spirulina platensis*, while the treatment groups were supplemented with 2.5 g/kg, 5 g/kg, and 10 g/kg of *Spirulina platensis*.

**Figure 2 vetsci-11-00383-f002:**
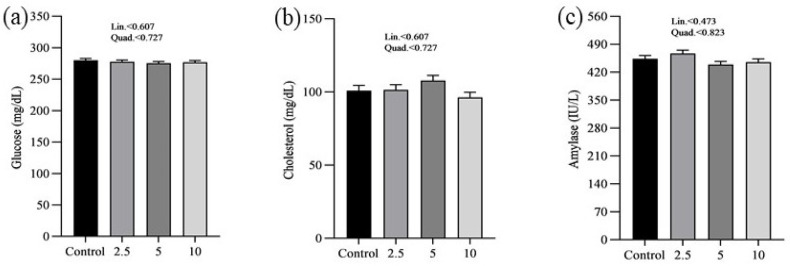
Effect of *Spirulina platensis* supplementation on serum nutritional biomarkers in laying hens. Values are presented as mean ± SEM for each group (n = 16). Panels illustrate the levels of different serum nutritional biomarkers: (**a**) glucose (mg/dL), (**b**) cholesterol (mg/dL), and (**c**) amylase (IU/L). The control group received no *Spirulina platensis*, while the treatment groups were given 2.5 g/kg, 5 g/kg, and 10 g/kg of *Spirulina platensis*.

**Figure 3 vetsci-11-00383-f003:**
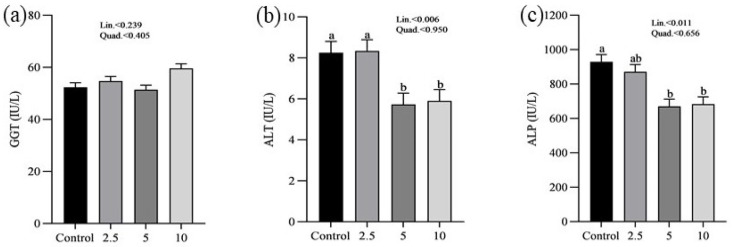
Effect of *Spirulina platensis* supplementation on liver enzymes in laying hens. Values are presented as mean ± standard error of the mean (SEM) for each group (n = 16). Different letters (a, b) above the bars indicate significant differences between groups (*p* < 0.05). (**a**) shows gamma-glutamyl transferase (GGT) levels, (**b**) shows alanine aminotransferase (ALT) levels, and (**c**) shows alkaline phosphatase (ALP) levels. The control group received no *Spirulina platensis*, while the treatment groups received 2.5 g/kg, 5 g/kg, and 10 g/kg of *Spirulina platensis*.

**Figure 4 vetsci-11-00383-f004:**
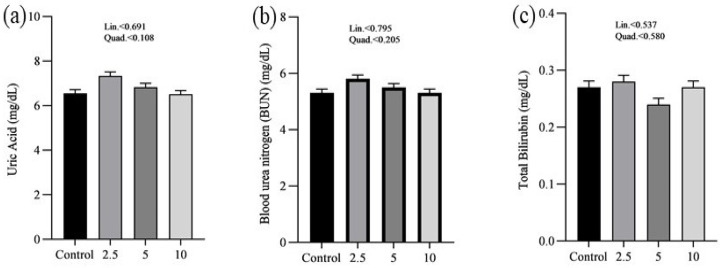
Effect of *Spirulina platensis* supplementation on kidney function markers in laying hens. Values are expressed as mean ± SEM for each group (n = 16). Panels represent different kidney function markers: (**a**) uric acid (mg/dL), (**b**) blood urea nitrogen (BUN) (mg/dL), and (**c**) total bilirubin (mg/dL). The control group received no *Spirulina platensis*, while the treatment groups were administered 2.5 g/kg, 5 g/kg, and 10 g/kg of *Spirulina platensis*.

**Figure 5 vetsci-11-00383-f005:**
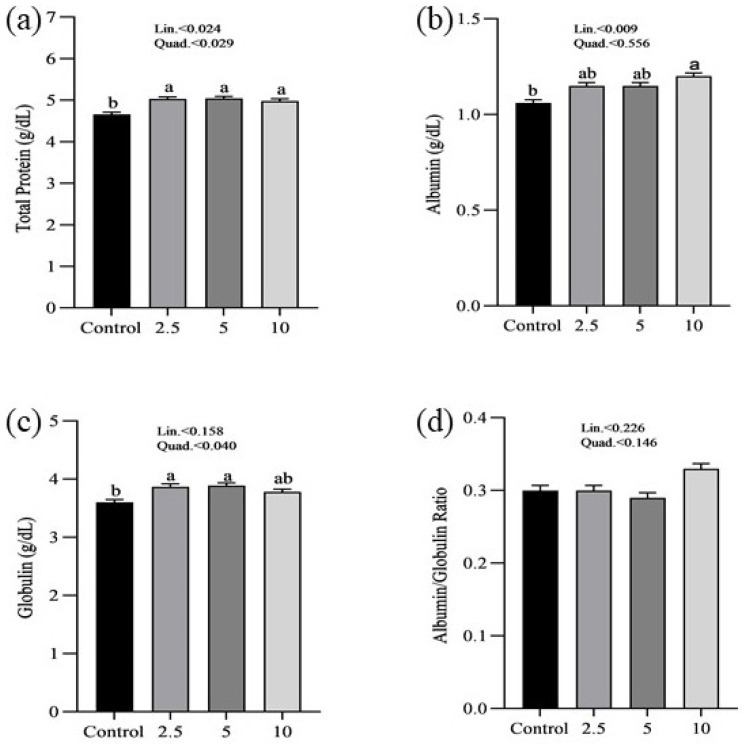
Effect of *Spirulina platensis* supplementation on protein metabolism in laying hens. Values are expressed as mean ± SEM for each group (n = 16). Panels show the levels of various protein metabolism indicators: (**a**) total protein (g/dL), (**b**) albumin (g/dL), (**c**) globulin (g/dL), and (**d**) albumin/globulin ratio. Different letters (a, b) indicate significant differences (*p* < 0.05) among groups. The control group received no *Spirulina platensis*, while the treatment groups were supplemented with 2.5 g/kg, 5 g/kg, and 10 g/kg of *Spirulina platensis*.

**Table 1 vetsci-11-00383-t001:** Ingredient composition and chemical analysis of laying hens’ basal diet.

Ingredients	%
Corn	63.12
Soybean meal 48% CP	22.56
Soybean oil	2.00
Vitamins premix ^1^	0.25
Minerals premix ^1^	0.05
Limestone	9.66
Mono-calcium phosphate	1.74
Salt	0.29
Sodium Bicarbonate	0.13
DL-Methionine	0.20
Total	100.00
Calculated composition	
Crude Protein (%)	16.50
ME (kcal/kg)	2882
Calcium (%)	4.00
Available P (%)	0.47
Crude fiber (%)	1.88
Crude fat (%)	4.41
Lysine (%)	0.84
Methionine (%)	0.46
Methionine + Cysteine (%)	0.74
Threonine (%)	0.61
Tryptophan (%)	0.21
Arginine (%)	1.04

CP: Crude protein; ME: Metabolizable energy; P: Phosphorus. ^1^ Vitamins–minerals premix provided in the following quantities per kilogram: 8,818,342 IU of vitamin A; 3,086,420 IU of vitamin D3; 36,742 IU of vitamin E; 13 mg of vitamin B12; 1177 mg of menadione; 4775 mg of riboflavin; 16,168 mg of d-pantothenic acid; 2350 mg of thiamine; 36,742 mg of niacin; 5732 mg of vitamin B6; 1398 mg of folic acid; 104,460 mg of choline; 441 mg of biotin; 120 mg of manganese (Mn); 1.4 mg of copper (Cu); 120 mg of zinc (Zn); 120 mg of iron (Fe); 0.5 mg of selenium (Se); and 800 mg of iodine (I).

**Table 2 vetsci-11-00383-t002:** The effect of *Spirulina platensis* on the productive performance of laying hens.

Parameters	*Spirulina platensis* (g/kg)				SEM	*p*-Value	
	0	2.5	5.0	10		Lin	Quad
39–40 weeks of age							
Egg weight (g/hen/day)	64.6	65.5	65.4	64.3	1.0	0.837	0.340
Egg mass (g/hen/day)	58.9	61.2	61.6	59.9	1.3	0.528	0.122
Hen day egg production (%)	91.2	93.6	94.3	93.1	1.6	0.381	0.287
Feed intake (g/hen/day)	114.4	120.0	116.8	119.9	1.7	0.091	0.469
Feed conversion ratio	1.948	1.964	1.895	2.010	0.039	0.494	0.203
41–42 weeks of age							
Egg weight, (g/hen/day)	64.1 ^b^	66.2 ^a^	66.4 ^a^	64.8 ^b^	0.8	0.531	0.023
Egg mass (g/hen/day)	58.9 ^b^	61.9 ^a^	62.6 ^a^	60.9 ^ab^	0.9	0.107	0.013
Hen day egg production (%)	92	93.4	94.3	94	1.6	0.319	0.575
Feed intake (g/hen/day)	120.3	118.9	119.2	117.1	3.0	0.498	0.915
Feed conversion ratio	2.054	1.924	1.902	1.926	0.057	0.143	0.212
43–44 weeks of age							
Egg weight, (g/hen/day)	63.6 ^b^	65.6 ^a^	66.3 ^a^	64.2 ^b^	1.0	0.603	0.045
Egg mass (g/hen/day)	57 ^b^	61.2 ^a^	62.3 ^a^	59.4 ^ab^	1.2	0.130	0.006
Hen day egg production (%)	89.6	93.3	94	92.9	2.0	0.242	0.224
Feed intake (g/hen/day)	110.6 ^b^	117.8 ^a^	117.5 ^a^	116 ^a^	1.7	0.041	0.015
Feed conversion ratio	1.946	1.925	1.886	1.958	0.034	0.978	0.202
39–44 weeks of age							
Egg weight (g/hen/day)	64.1 ^b^	65.8 ^a^	66.1 ^a^	64.4 ^b^	0.8	0.732	0.042
Egg mass (g/hen/day)	58.2 ^b^	61.4 ^a^	62.2 ^a^	60.1 ^ab^	0.9	0.120	0.005
Hen day egg production (%)	90.9	93.4	94.2	93.3	1.4	0.216	0.241
Feed intake (g/hen/day)	115.1	118.9	117.8	117.7	1.5	0.314	0.190
Feed conversion ratio	1.982	1.937	1.894	1.959	0.033	0.453	0.099

Means not sharing a common letter (^a–b^) in a row are significantly different (*p* < 0.05). SEM: standard error of the means; Lin: linear responses to dietary inclusion levels; Quad: quadratic responses to dietary inclusion levels.

**Table 3 vetsci-11-00383-t003:** The effect of *Spirulina platensis* on egg quality criteria of laying hens.

Parameters	*Spirulina platensis* (g/kg)				SEM	*p*-Value	
	0	2.5	5.0	10		Lin	Quad
40 weeks of age							
Haugh unit score	73.12 ^b^	78.04 ^ab^	82.98 ^a^	74.52 ^ab^	1.47	0.485	0.022
Yolk color score	3.92 ^c^	4.60 ^b^	5.26 ^a^	5.07 ^ab^	0.12	<0.001	0.036
Egg strength (kg)	5.66	5.73	5.35	5.82	0.14	0.939	0.491
Shell thickness (mm)	0.63	0.58	0.62	0.57	0.01	0.250	0.771
42 weeks of age							
Haugh unit score	75.08	74.11	72.58	76.63	2.22	0.755	0.261
Yolk color score	3.62 ^b^	3.95 ^b^	4.79 ^a^	4.70 ^a^	0.13	<0.001	0.122
Egg strength (kg)	5.57	5.95	6.01	5.91	0.21	0.261	0.268
Shell thickness (mm)	0.47	0.47	0.45	0.47	0.01	0.550	0.533
44 weeks of age							
Haugh unit score	72.98	76.08	78.22	71.29	3.19	0.837	0.119
Yolk color score	3.79 ^b^	4.20 ^a^	4.29 ^a^	4.50 ^a^	0.12	<0.001	0.395
Egg strength (kg)	4.76 ^b^	5.36 ^a^	5.39 ^a^	5.04 ^a^	0.21	0.374	0.032
Shell thickness (mm)	0.41	0.42	0.43	0.42	0.01	0.610	0.304
Shell (%)	13.55 ^a^	13.13 ^b^	13.01 ^b^	13.60 ^a^	0.22	0.980	0.028
Albumen (%)	52.71	54.59	53.37	53.53	0.48	0.560	0.077
Yolk (%)	33.73	32.27	33.60	32.86	0.43	0.509	0.412

Means not sharing a common letter (^a–b^) in a row are significantly different (*p* < 0.05). SEM: standard error of the means; Lin: linear responses to dietary inclusion levels; Quad: quadratic responses to dietary inclusion levels.

**Table 4 vetsci-11-00383-t004:** The effect of *Spirulina platensis* on egg quality criteria of storage eggs.

Parameters	*Spirulina platensis* (g/kg)				SEM	*p*-Value	
	0	2.5	5.0	10		Lin	Quad
Egg weight (g)	61.95	63.91	61.85	61.66	0.42	0.430	0.193
Haugh unit score	68.32 ^b^	68.00 ^b^	79.13 ^a^	73.13 ^ab^	1.52	0.050	0.337
Yolk color score	4.00 ^b^	4.41 ^a^	4.50 ^a^	4.46 ^a^	0.06	0.009	0.067
Egg strength (kg)	5.35 ^b^	5.94 ^a^	6.22 ^a^	6.04 ^a^	0.09	0.005	0.035
Shell thickness (mm)	0.40 ^b^	0.42 ^a^	0.43 ^a^	0.41 ^ab^	0.003	0.102	0.006
Shell (%)	12.63 ^b^	12.90 ^ab^	13.15 ^ab^	13.32 ^a^	0.10	0.010	0.781
Albumen (%)	50.49 ^a^	50.26 ^a^	49.24 ^ab^	48.09 ^b^	0.32	0.003	0.464
Yolk (%)	36.87 ^b^	36.84 ^b^	37.61 ^ab^	38.59 ^a^	0.28	0.019	0.359

Means not sharing a common letter (^a–b^) in a row are significantly different (*p* < 0.05). SEM: standard error of the means; Lin: linear responses to dietary inclusion levels; Quad: quadratic responses to dietary inclusion levels.

## Data Availability

The supporting data of this study are available within the article.

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
