# Peer review of "Enhancing Laying Hens’ Performance, Egg Quality, Shelf Life during Storage, and Blood Biochemistry with Spirulina platensis Supplementation"

_vetsci, 2024, doi:10.3390/vetsci11080383_

Round 1
Reviewer 1 Report
Comments and Suggestions for Authors
It's not clear how long the adaptation diet was fed, and the age of the hens when the trial started. Otherwise, the implication is that the first reported week in the results was actually the adaptation period.
Experimental diets seem simple, practical and at least-cost. What additional cost did SP supplementation impose?
How much blood was collected from each hen? Was egg quality assessed in the first week of collection?
What could be the reason for similar responses observed in 0% and 10% SP supplementation levels, respectively? Any speculations?
More comments in the document.

Author Response
Dear Reviewer,
Thank you for dedicating your time and effort to review our manuscript. We have addressed your comments and revised the manuscript accordingly. Please find our responses and the revised manuscript in the attached Word file.

Reviewer 2 Report
Comments and Suggestions for Authors
INTRODUCTION
Lines 44 to 50 - The phrase “Poultry farming is crucial for supplying the global demand for animal protein, and enhancing the health and efficiency of laying hens is essential for the business’s long‐term sustainability and effectiveness. Conventional feed ingredients mainly focus on fulfilling the nutritional requirements of hens [1]. Nevertheless, there is an increasing interest in using functional feed additives that can improve health and performance as well as the quality of their end products, such as eggs.” is absolutely unnecessary. You can join with the other paragraphs or simply delete it.
Lines 51 to 57 - please resume the phrase below in a small phrase or 2 or 3 lines.
“In recent years, there has been a growing interest in utilizing algae to enhance the performance and health of poultry. Algae are an array of photosynthetic organisms that inhabited aquatic environments such as freshwater and marine ecosystems [2]. Algae have a wide range of sizes and degrees of intricacy, spanning from single‐celled tiny organisms to large multicellular seaweeds. There are about 100,000 species of microalgae that can be categorized into four distinct groups: eukaryotic diatoms, green algae, golden algae, and blue‐green algae [3]”
You need to improve the introduction to give more specific information about SP in a practical view in poultry nutrition and health. You must explain the reasons why the results were obtained. It is not informing that performance or health were improved but why they were improved and justify your experiment. This is not clear in your text.
There some questions that are not included in your introduction:
1. The idea is using SP as an ingredient or additive ? What is the function ?
2. The bioactive compounds are not described.
3. Why did you proposed the inclusion rates of 2,5, 5 and 10 g/kg ?
4. Are the inclusion rates proposed for an additive/functional/nutritional purpose ?
MATERIALS AND METHODS
Table 1 – the undertable information need to be corrected formatted
Why did you not show selenium content in the diet
Line 105 - The experiment was carried out for a six‐week duration. Why did you run the study for just 6 weeks ? Is this time enough to observe any important results ? Did you submit hens to an adaptation period ? I believe this is a short time for any kind of response.
Lines 133 and 134 – please, why did you not analyzed kg of feed intake per dozen eggs ?
Lines 160 to 164
You have listed various serum components such as sodium, chloride, calcium, phosphorus, glucose, cholesterol, amylase, uric acid, blood urea nitrogen (BUN), total bilirubin, alanine aminotransferase (ALT), gamma‐glutamyl transferase (GGT), alkaline phosphatase (ALP), total protein, albumin and globulin.
In the item 2.5. Statistical analysis, you described that you used Duncan’s multiple range test to analyze treatments in treatments formed by dosis. Why did you not test regression models, especially, the polynomial ones ?
RESULTS
Table 2
The variables Egg weight (g/hen/day), Egg mass (g/hen/day), Hen Day egg production, (%), and Feed intake (g/hen/day) do not need to be presented with 2 decimals, just one is enough. For Feed conversion ratio must be presented with 3 decimals.
A total period performance evaluation is the most important information to be presented (39-45 wk). There is a carry-on effect from one phase to another. Evaluate two-weeks periods is not enough to obtain an important conclusion about the use of SP in hen’s diets.
In figure 2, you considered Glucose, Cholesterol and Amylase as nutritional biomarkers ? Why ? I believe they are biochemical compounds and cannot use alone as metabolic biomarkers.
DISCUSSION
Lines 341 to 344 -In the phrase: “The results of the present study demonstrate that the incorporation of SP to the diet of laying hens has diverse impacts on their productivity, egg quality, egg shelf life and serum biochemical parameters. These effects differ depending on the levels of the SP supplementation in the diets.” I believe that this interpretation of the results is not coincidental. In my opinion, there are some reasons:
- You analyzed data for a short period (39 – 45 weeks).
- The polynomial regression was not included in the statistical analysis.
- You should consider a multivariate analysis to observe the most relevant results.
- I believe you could be more effective in the scientific basis for SP. It is difficult to understand if you tested as an ingredient or additive/functional ingredient.
It is important to emphasize that this experiment was correctly designed but must be extended to more weeks (at least more 10 weeks) and analyzed by regression, multivariate data and your conclusion is based in data that must be improved. The scientific basis is not sufficient to support the conclusion.
Author Response

(The authors gave the same response as above.)

Reviewer 3 Report
Comments and Suggestions for Authors
The objective of this MS was to study the effect of different SP addition level in feed on growth performance, egg quality, egg storage time, and blood biochemical indices of chickens. The experimental design is complete and informative. However, there are some comments need to be addressed.
General comments
1. In the introduction part, the authors wrote there were different articles made different conclusion by using different SP level. It should be detailed and compared in the discussion part.
2. The conclusion of the abstract and the main text is different: the former showed the suitable addition level is 10 g/kg, while the latter showed 2.5 and 5 g/kg. Please make consistent.
3. There are some English grammatical errors and punctuation errors, which the MS difficult to read.
Specific comments
1. The abbreviation of SP should be noted at the first time appeared (L57).
2. L80 the sentence should be reorganized.
3. The source of the SP should be specified.
4. There are errors in Figures 4 and 5. Inconsistency between images and content.
Comments on the Quality of English Language
need to be revised.
Author Response

(The authors gave the same response as above.)

Reviewer 4 Report
Comments and Suggestions for Authors
Enhancing Laying Hens Performance, Egg Quality and Shelf Life during Storage, and Blood Biochemistry with Spirulina platensis Supplementation
Review Response:
L 18: What is meant by ‘higher’ yolk color?
L 23: Promote improved health.. Redundant use of words
L 40: Productivity of what?
L 67: what kind of utilization of SP? Dietary?
L67-84: It needs to strengthen the justification of research. In particular, how is the current research different from previous studies on supplementing spirulina in layer hen diets? What are inclusion levels tested so far? What is the basis for selecting the tested inclusion levels in the current study?
L 83-84: These observed variations in the impact of SP on laying hens may be attributed to the usage of appropriate amounts in their diets. Very subjective and not clear!
L 70: El‐Hady et al. [15] finding, the observed improvement in broiler chickens, compared to what? What was replaced by 6% SP in the broiler chicken diet?
L 89-90: Achieving this will contribute to the sustainability and efficiency of laying hen’s production… As an objective, it is too broad and it is not clear how this objective is exactly achieved.
Table 1: Provide the ingredient composition and calculated analysis as well. It is very important addition needed in a revised version. Therefore, I am suggesting a major revision. Because, based on the final diet composition, the discussion can be changed.
L 132: The equation for egg mass is not clear. What are the units? Why are two parameters separately analysed as egg weight and egg mass? What is the difference?
L 140-141: Not clear, biweekly and three weeks post experiment… is confusing
L 144-146: Elaborate on how the egg quality parameters were measured. Units are confusing in the results.
Follow the journal guidelines for the table editing.
L 345-365: How would you discuss the reduced egg weight and egg mass in birds given 10 g/kg SP during 43-45 weeks of age? In the abstract, SP supplementation up to 10 g/kg is recommended. After seeing this result, it is now doubtful.
Conclusion: What is the optimum inclusion level of SP according to the findings? What is the basis of your recommendation?
Comments on the Quality of English LanguageVery subjective use of adjectives is noted.
L 427: Dramatically improve..
L 452: Eggs resilience...?
Some sentences are started with an abbreviation.
A more specific approach in scientific writing is recommended. Grammatical errors are minimal.
Author Response

(The authors gave the same response as above.)
